# Seasonal and inter-annual variations in carbon fluxes in a tropical river system (Tana River, Kenya)

Naomi Geeraert<sup>1</sup>, Fred O. Omengo<sup>1,2</sup>, Fredrick Tamooh<sup>1,3</sup>, Trent R. Marwick<sup>1</sup>, Alberto V. Borges<sup>4</sup>, Gerard Govers<sup>1</sup>, Steven Bouillon<sup>1</sup>

<sup>1</sup>KU Leuven, Department of Earth and Environmental Sciences, Celestijnenlaan 200E, 3001 Leuven, Belgium
 <sup>2</sup>Kenya Wildlife Service, P.O. Box 40241-00100, Nairobi, Kenya
 <sup>3</sup>Kenyatta University, Department of Zoological Sciences, P.O Box 16778-80100, Mombasa, Kenya
 <sup>4</sup>Université de Liège, Unité d'Océanographie Chimique, Institut de Physique (B5), 4000 Liège, Belgium

Correspondence to: Naomi Geeraert (naomi.geeraert@kuleuven.be)

- Abstract. Quantification of sediment and carbon (C) fluxes in rivers with strong seasonal and inter-annual variability presents a challenge for global flux estimates as measurement periods are often too short to cover all hydrological conditions. We studied the dynamics of the Tana River (Kenya) from 2012 to 2014 through daily monitoring of sediment concentrations at three sites (Garissa, Tana River Primate Reserve and Garsen) and daily monitoring of C concentrations in Garissa and Garsen during three distinct seasons. In wet seasons, C fluxes were dominated by particulate organic C (POC)
- and decreased downstream. Dry season fluxes of dissolved inorganic C (DIC) and POC had a similar share in total C flux at both locations while POC fluxes increased downstream. The dissolved organic C (DOC) flux did not show strong spatial nor temporal variations. The construction of constituent rating curves with a bootstrap method in combination with daily discharge data (1942-2014) provided potential sediment and C flux ranges as a function of annual discharge. At low annual discharge, our estimates generally predict a net decrease of sediment and C storage between the upstream and downstream
- site. As the annual discharge increases, our simulations shift toward net retention. This analysis allowed us to infer how variations in discharge regime, related to climate or human impacts, may affect riverine fluxes. Overall, we estimate that retention was dominant: integration over all simulations resulted in an average net retention of sediment (~2.9 Mt yr<sup>-1</sup>), POC (~18000 tC yr<sup>-1</sup>), DOC (~920 tC yr<sup>-1</sup>) and DIC (~1200 tC yr<sup>-1</sup>) over the 73 years of discharge measurements.

#### **1** Introduction

- The riverine flux of carbon (C) to the ocean is an important component of the global C budget, and of interest both from a terrestrial perspective (representing a loss of terrestrial organic C), and from a marine perspective (representing an input of organic and inorganic C). Efforts to constrain these fluxes were initiated several decades ago, and either established relationships between the total annual river flow and the total organic carbon (TOC) load or calculated the loss of organic carbon (OC) based on the spatial extent of terrestrial ecosystems and their expected areal rate of C loss (Degens et al., 1982, 1981).
- 1991; Garrels and Mackenzie, 1971; Meybeck, 1987; Schlesinger and Melack, 1981). However, few empirical data on C

loads existed at that point. As more data became available, increasingly refined models incorporated global spatial data-sets of precipitation, temperature and vegetation (Ludwig et al., 1996), the most recent being the Global Nutrient Export from Watersheds (NEWS and NEWS2) models, which include relationships with ecological, climatic and geomorphologic characteristics of the catchment (Mayorga et al., 2010; Seitzinger et al., 2005). In recent models, the particulate OC (POC)

- flux is often estimated based on the sediment flux and the %OC in the solid flux (Beusen et al., 2005; Ludwig et al., 1996), while the dissolved OC (DOC) flux is estimated based on relationships with soil characteristics in the catchment, either through the C/N ratio or through the C content and vegetation cover (Aitkenhead and McDowell, 2000; Ludwig et al., 1996). Inorganic C (IC) fluxes are estimated based on the variation of weathering and erosion-related parameters between and within catchments (Ludwig et al., 1998).
- The robustness of these global estimates strongly depends on the reliability and representativeness of the underlying datasets. The historical paucity of datasets representing tropical river systems has been a strong impetus for the burgeoning research on sediment and C fluxes within these systems over the past decade, and has also witnessed a broadening geographical coverage: while the Amazon has traditionally functioned as the archetype for tropical rivers (Abril et al., 2014; Mayorga et al., 2005; Moreira-Turcq et al., 2003, 2013; Richey et al., 2002), there has been an increase in data availability from other
- systems in Latin-America (Depetris and Kempe, 1993; Laraque et al., 2013; Mora et al., 2014), as well as a range of systems in Asia (Aldrian et al., 2008; Bird et al., 2008; Sarin et al., 2002; Zhou et al., 2013), small islands (Lloret et al., 2011; Wiegner et al., 2009), and Africa (Borges et al., 2015; Bouillon et al., 2012; Brunet et al., 2009; Coynel et al., 2005; Tamooh et al., 2014; Wang et al., 2013; Zurbrügg et al., 2013).

Although the spatial extent of river basins where C dynamics have been studied is expanding, the temporal resolution is

- often relatively low, with many studies based on a limited number of longitudinal cruises along the river. Monitoring campaigns most often achieve only monthly measurements or, in a few cases, 2-weekly or weekly measurements (e.g. Bouillon et al., 2012; Brunet et al., 2009; Tamooh et al. 2014). The monitoring period in most cases lasted one or two years. However, tropical rivers are often characterized by a strong seasonality and a high inter-annual variability (Syvitski et al., 2014), of which the range is difficult to capture with either longitudinal or relatively short monitoring campaigns.
- To assess the effect of sampling frequency on the annual sediment and C flux estimates of a tropical river system, we focused on the lower Tana River (Kenya) with measurements at Garissa (GSA), Tana River Primate Reserve (TRPR) and Garsen (GSN) (Figure 1). A newly collected dataset was merged with available data on sediment and C for the lower Tana River. The previously available data originate from longitudinal surveys (Bouillon et al., 2009; Tamooh et al., 2012, 2013) and monthly sampling efforts at Garissa and Tana River Primate Reserve (Tamooh et al., 2014). Based on those datasets, a
- strong retention of sediment, POC and DOC was inferred, which was then related to the exchange of C between the river and its floodplains. However, even a sampling periodicity of two weeks was not able to capture the high variability in discharge, with peaks lasting for ca. 5 days, and the response of the C concentration during the wet seasons. Therefore, a newly collected dataset focused on the daily C dynamics during three wet seasons at Garissa and Garsen (Geeraert et al., 2017).

Additionally, suspended sediment samples were taken at three sites (Garissa, TRPR and Garsen) at a daily frequency (less at Garissa due to logistical constraints) for almost 2 years.

This extensive dataset allowed us to assess how different hydrological conditions affected sediment and C fluxes. Furthermore, we examined to what extent an increase in sampling frequency improved the accuracy of flux estimates by calculating annual fluxes based on daily to monthly samples. Applying a bootstrap method during the construction of the constituent rating curves provided insight in the robustness of those estimates. By using historical discharge data as representative samples of discharge variations, the calculated annual fluxes revealed the relationship between retention or mobilization between the sites as a function of the annual discharge. This relationship was ultimately used to assess the impact of hydrological changes, either due to climate change or due to the construction of additional dams in the upper

catchment, on sediment and C retention and mobilisation.

## 2 Methodology

#### 2.1 Study area

The Tana River, with a length of ca. 1100km, is the longest river in Kenya (Figure 1). It originates in the highlands of the Aberdare Mountain Range and Mount Kenya. Five major reservoirs have been built in the upper reaches of the catchment of

- which Masinga Reservoir is the largest (~1.56 km<sup>3</sup>). A significantly larger dam, the High Grand Falls (HGF) dam (5.7 km<sup>3</sup>), is planned downstream of the existing dams (Tana and Athi Rivers Development Authority, 2016). Approximately 600 km from the river mouth, the river enters a semi-arid environment, where it starts meandering through a forested floodplain seldom wider than 5 km. Beyond the floodplain, the landscape is characterized by open savannah vegetation. The last 70 km before reaching the Indian Ocean, a deltaic system with mangrove forests and seasonally flooded grasslands is encountered.
- The focus of this research is the lower reach of the Tana River, with sampling sites at Garissa, Tana River Primate Reserve and Garsen, measured along the river at 455 km, 155 km and 70 km from the river mouth (Figure 1). No permanent tributaries are present over this distance, which makes it an ideal setting to compare incoming and outgoing fluxes of water, sediment and C (Geeraert et al., 2017). Some "lagas" (temporary rivers) are present, but they are only active for a few days per year.
- Precipitation is concentrated within the upper catchment where the average annual rainfall exceeds 1500 mm yr<sup>-1</sup>, whereas the average annual precipitation near Garissa is less than 350 mm yr<sup>-1</sup>. The majority of rainfall occurs across two wet seasons; the long wet season spanning March through May, and the short wet season which takes place from October to December. Besides the spatial and seasonal variability, the inter-annual variability in the amount of precipitation and the onset and ending of the wet seasons is very high (Indeje et al., 2000).

25

# 2.2 Long-term discharge dataset

Daily water levels were recorded by government institutions (nowadays Water Resource Management Authority, WRMA) at Garissa and are available either as discharge (from 18<sup>th</sup> of April, 1941 until January, 2012) or as gauge height (from 1<sup>st</sup> of April, 2010 until 31<sup>st</sup> December 2014). A discharge rating curve based on 19 manual measurements (SEBA F1 Universal

5 current meter) by WRMA between 2007 and 2011 and 11 measurements with an Acoustic Doppler Current Profiler (ADCP, Teledyne RiverRay, our data) in 2012 and 2013, was constructed to convert the recent measurements from gauge height to discharge (Figure 2a).

Gaps in the daily discharge series (1754 missing days out of 26 952 days) were filled by taking the average value of the discharge at that date (Julian day) over all the observation years. This simple method is suitable for our purpose as we will

use the discharge dataset as a realistic hydrological input for a whole range of annual discharge patterns and not to reconstruct historical fluxes related to a specific year.
 The discharge dataset of Garsen covers the period between September 1950 and November 2014. However, there were large

The discharge dataset of Garsen covers the period between September 1950 and November 2014. However, there were large inconsistencies between different subseries which were likely related to the fact that gauge boards were installed at different locations over the data collection period: the maximum discharge in the dataset without gauge heights was less than 300 m<sup>3</sup>

- s<sup>-1</sup>, while present-day discharge measurements with an ADCP (Teledyne RiverRay) ranged up to 450 m<sup>3</sup> s<sup>-1</sup> in the main channel. To maintain consistency in the dataset for the flux reconstructions since 1941, the regression equation between the discharge in Garissa and the discharge in Garsen is based on the period for which gauge heights were available (24<sup>th</sup> of October, 2010 until 25<sup>th</sup> of November, 2014). The conversion from gauge heights to discharge was based on 28 discharge measurements with the ADCP in 2012 and 2013 (Figure 2b).
- The regression equation between the discharge in Garissa and Garsen was biphasic according to the discharge in Garissa with a breakpoint at 550 m<sup>3</sup> s<sup>-1</sup>. Under non-flooded discharge conditions, the travel time of the water was approximately 5 days. Above 550 m<sup>3</sup> s<sup>-1</sup>, there was no consistent relationship between the upstream and downstream discharge due to the longer and varying water retention time caused by extensive flooding as well as the additional inflow from the floodplain to the river during the falling stage. Therefore, a constant discharge value was used during high discharges in Garissa:
  - $Q_{GSN}(i) = 32.2665 + 0.6456 Q_{GSA}(i-5) \qquad \text{for } Q_{GSA}(i-5) < 550 \text{ m}^3 \text{ s}^{-1}$  $Q_{GSN}(i) = 387 \text{m}^3 \text{s}^{-1} \qquad \text{for } Q_{GSA}(i-5) > 550 \text{ m}^3 \text{ s}^{-1}$

with  $Q_{GSN}$  and  $Q_{GSA}$  being the discharge in m<sup>3</sup> s<sup>-1</sup> at Garsen and Garissa and *i* being the date.

There were no discharge observations available at TRPR. Therefore, we calculated the discharge at TRPR as a distanceweighted average between Garissa and Garsen.

Relationships between discharge and concentrations of suspended sediments and different C pools were expected to be dependent on flow conditions (non-flooded and flooded). Therefore, we defined the start of a flooded period at the third day that the discharge at Garissa exceeded 550 m<sup>3</sup> s<sup>-1</sup>, provided that the total duration of the period for which this threshold value was exceeded was at least 5 days (consistent with the criteria in Geeraert et al. (2015)). We assumed that the flooded state

persisted until the discharge at Garissa dropped below 150 m<sup>3</sup> s<sup>-1</sup>. Flooding in TRPR and Garsen was assumed to occur with a delay of 4 and 5 days with respect to Garissa, respectively.

## 2.3 Fluxes of sediment and carbon

- Riverine fluxes of suspended sediment and different C pools were calculated using a combination of the different datasets.
  Data on total suspended matter (TSM) concentrations were gathered during (1) monthly or biweekly monitoring between January 2009 and December 2013 at Garissa and between September 2011 and December 2013 at TRPR, (2) daily sampling in Garissa, TRPR and Garsen between May 2012 and March 2014, and (3) three wet season campaigns conducted in Garissa and Garsen in 2012, 2013 and 2014. The sampling protocol for TSM concentrations is further explained in (Geeraert et al., 2015).
- TSM concentrations for non-sampled days were calculated in R based on non-linear least-squares regression equations between the untransformed discharge and TSM, whereby a distinction was made between non-flooded and flooded conditions, according to the previously discussed thresholds. Suspended sediment transport was related to discharge through a power function,  $TSM=a.Q^b$ , whereby *TSM* denotes the suspended sediment concentration in mg L<sup>-1</sup>, *Q* the discharge in m<sup>3</sup> s<sup>-1</sup> while *a* and *b* are regression coefficients (Figure 3, Table 1). Daily sediment fluxes were obtained by multiplication of the
- daily concentration with the daily discharge, and summing over each year resulted in the annual sediment flux. Data on concentrations of POC, DOC, and dissolved IC (DIC) were similarly combined from different datasets. Daily POC, DOC and DIC concentration data at Garissa and Garsen were obtained during three wet season campaigns in 2012 (n=37 per site), 2013 (n=52 per site) and 2014 (n=45 per site) (Geeraert et al., 2017). The dataset of Garissa was further expanded with data from monthly (Tamooh et al., 2014) and biweekly (unpublished data) monitoring over the period January 2009 -
- December 2013 (n<sub>POC</sub>=85, n<sub>DOC</sub>=68, n<sub>DIC</sub>=86). The sampling protocols used to obtain concentrations of POC, DOC and DIC have been presented in Geeraert et al. (2017) for the wet season campaigns, and in Tamooh et al. (2014) for the biweekly and monthly samples, where a different method for DIC concentration was used.

The annual fluxes of POC and DOC were calculated in R based on nonlinear least-squares regression equations between the untransformed C concentration and the discharge (Figure 4). The regression coefficients for POC and DOC were calculated

- for the equation  $C=a.Q^b$  whereby *C* denotes the C concentration in mg L<sup>-1</sup>. Due to the absence of a satisfactory regression model for the non-flooded concentrations of DIC, the DIC concentration was estimated based on a single linear regression curve (C=aQ+b) at each site, including both the non-flooded and flooded observations. A maximum POC concentration of 125 mg L<sup>-1</sup> was set for both sites to avoid an unrealistic overestimation during non-flooded conditions. Daily C concentrations were multiplied with daily discharge and the sum per year was made for each of the C species.
- The uncertainty on the regression curves of TSM and the C species was simulated by a bootstrap method. By using the package 'Boot' in R (version 1.3-18), a subset of the original dataset was constructed by randomly omitting certain observations and selecting other observations multiple times in such a way that the initial number of observations is maintained. Regression coefficients were calculated for 200 of those subsets. A maximum limit was set for the daily

concentrations of POC (125 mg  $L^{-1}$ ) and TSM (1000 mg  $L^{-1}$ ), as some regression lines resulted in unrealistically high values at high, but not yet flooded, discharge conditions. Subsequently, the annual fluxes were calculated from the daily concentrations from each regression line, or by a combination of regression lines for the species where the distinction between flooded and non-flooded conditions is made.

# 5 2.4 Analysis of seasonal variation

To examine seasonal patterns in sediment and C fluxes, we focus on the years 2012 and 2013 since this period has the most dense data availability for TSM, POC, DOC and DIC, and the gaps in the daily discharge dataset during this period (135 days in Garissa, 9 days in Garsen) occurred mainly during dry conditions, where data filling is relatively straightforward and does not influence the annual flux estimates substantially. The delineation of the season was done by fixed dates in order to

10 have the same number of days per season for each year. The long wet season ran from April 1<sup>st</sup> until June 15<sup>th</sup> (n=76 days), while the short wet season ran from October 15<sup>th</sup> until December 31<sup>st</sup> (n=78 days). Together, the two remaining periods of the year formed the dry season (n=212 days in 2012 and 211 days in 2013).

## 2.5 Analysis of sampling frequency

To assess the influence of the sampling frequency on the annual TSM flux, we used the TSM measurements of Garsen. A 15 total of 644 samples were collected at this site over 657 days (May 2012 - March 2014), of which 55 samples were taken under flooded conditions: the latter were all taken in 2013 as no flooding occurred in 2012. Subsets of the total dataset were taken at regular intervals at a frequency of 1, 2, 3, 4, 7, 14, 21 and 30 days. Up to four different starting dates were chosen per interval in order to assess the variability within a certain sampling interval, resulting in a total of 26 subsets.

Three regression curves were fit to each subset: one with all data included and two where the distinction was made between flooded and non-flooded conditions according to the previously discussed criteria. The scenario with all data included simulates the situation where there is no prior knowledge about the existence of different hydrological conditions. No distinction according to conditions was made for a sampling frequency of 30 days and for one subset of 21 days because there were insufficient observations during flooded conditions. Subsequently, the annual fluxes were calculated for 2012 and 2013 by applying the regression curves to the daily water discharge measurements and summing the daily sediment fluxes.

## 25 3 Results

## **3.1 Seasonal variations**

The total dry season discharge was very constant, both spatially (Garissa vs. Garsen) and temporally (2012 vs. 2013), with values of 1.9 km<sup>3</sup> and 2.1 km<sup>3</sup> in Garissa and 1.9 km<sup>3</sup> and 2.0 km<sup>3</sup> in Garsen, in 2012 and 2013 respectively (Figure 5a). The dry seasons still had a fair share in the total annual discharge (34-44%), given that their time spanned ~58% of the year. The

between the two sites. During each wet season, we observed a significant downstream decrease in discharge of ca. 25% in both seasons of 2012, and of 37% and 17% during the long and short wet seasons of 2013. This decrease is caused by infiltration in the river bed and/or floodplain, evaporation and the absence of considerable tributaries. The two wet seasons of 2012 had a comparable relative contribution to the annual water flux at each site (28-32%), while in 2013 the flooded long wet season took a significantly larger share (48% and 39%) in comparison to the suppressed short wet season (18% and

19%).

The dry seasons had a lower relative contribution in the annual flux of the sediment and total carbon (TC) than it was for the water flux (Figure 5b, Figure 5c). Furthermore, a clear downstream increase of the TC flux was observed during the dry season. The downstream increase in TC during the dry seasons in 2012 and 2013 was less than the decrease observed during

- the wet seasons, leading to a net downstream decrease in TC over the whole year. The relative contribution of the dry season to the total annual fluxes of water, sediment and C was always more important downstream in Garsen than it was upstream in Garissa, which illustrates that the floodplain downstream of Garissa acted as a buffer during the high flows by reducing and/or delaying fluxes. The large differences in the relative seasonal contribution of the water, sediment and C fluxes indicated that the daily averaged concentration of TSM and C varies strongly between the seasons (Figure 5).
- POC was found to be the dominant C pool on both the annual scale and during the wet seasons (Figure 6). During the dry season, the POC and DIC fluxes were similar in magnitude. The POC flux strongly increased downstream during the dry season and to a lesser extent during the short wet season of 2013, while during the long wet seasons, a decrease was observed. The DOC flux during the dry seasons was constant in space and time. During the non-flooded wet seasons, there was a slight downstream decrease in DOC flux, while there was no change during the flooded wet season in 2013. The DIC
- flux increased slightly during the dry season, while there was a limited decrease during all wet seasons.

## 3.2 Differences in longitudinal fluxes

The difference between the incoming flux at Garissa and the outgoing flux at Garsen determines whether the river stretch was either a retention area or a source area for the sediment or C with respect to the various riverine pools. While we use the term *retention* when the downstream flux was smaller than the upstream flux, it does not imply anything about the fate of the

- retained C, which can either be stored in the river or floodplain or can be further processed (including outgassing to the atmosphere in the case of DIC). The flux differences were calculated for random combinations of the 200 annual fluxes per year at each site and were plotted as a function of the total annual discharge at Garissa (Figure 7, left graphs). Similarly, the difference in fluxes was also expressed as a percentage of the upstream flux at Garissa (Figure 7, right panels).
- Retention of sediment occurred under all annual water fluxes, except where the water flux was <3 km<sup>3</sup> yr<sup>-1</sup>. Furthermore, a larger annual water flux led to a larger absolute deposition of sediment (Figure 7a). The POC flux difference exhibited a slightly different pattern: below a water flux of ca. 3.5 km<sup>3</sup> yr<sup>-1</sup> there was usually a net downstream increase in C flux, between 3.5 and 6 km<sup>3</sup> yr<sup>-1</sup> the C fluxes were quite well balanced, while above 6 km<sup>3</sup> yr<sup>-1</sup> a general retention of POC was estimated (Figure 7c). Both for TSM and POC, the relative change ranged between -106% and 78%. The pattern of DOC

was less consistent (Figure 7e-f). Initially, there was an increase in DOC flux difference between the sites with increasing discharge. However, once more days with flooded conditions were occurring, there was considerable scatter, whereby the negative differences (net export over the river stretch considered) occurred in years with a high number (>100) of flooded days. The relative change in DOC fluxes (-33% to 29%) had a smaller range than for POC. Finally, the DIC fluxes indicated

5 a larger downstream DIC flux than the upstream DIC flux at an annual discharge below ~5 km<sup>3</sup> yr<sup>-1</sup>, while at higher discharge, the upstream DIC flux became larger (Figure 7g-h). This is contrasting to what might be expected from the rating curves, but can be explained by the large downstream decrease in water discharge. As was the case for DOC, the relative change in DIC fluxes was limited (-29% to 27%).

The cumulative effect over all the annual net differences over the whole historical observation period resulted in a net

- average annual sediment deposition rate of ~2.9 Mt yr<sup>-1</sup>. This significant amount of sediment deposition implies that a large amount of unconsolidated sediment is available for autogenic processes to keep the sediment concentration near transport capacity (Geeraert et al., 2015). The long-term balance of the C species also suggests a net retention of C within the river stretch for POC, DOC and DIC, with an average annual amount of ~18 000 tC yr<sup>-1</sup>, ~920 tC yr<sup>-1</sup> and ~1200 tC yr<sup>-1</sup>, respectively. Although these measurements give no indication to the fate of the retained C, part of the POC will likely be
- deposited in the floodplain while some will be respired, while the excess DIC is expected to have evaded to the atmosphere, as has been quantified for distinct wet seasons in the Tana River (Geeraert et al., 2017).

#### 3.3 Carbon species differentiation as a function of discharge

The relative contribution of each C species in the TC flux was also correlated with the annual water flux (Figure 8). The contribution of POC in Garissa was relatively low (~ 30-40%) during dry years, but increased notably with increasing annual

- discharge and became stable around 60-70%. At Garsen, the contribution of POC was also relatively low at low discharge, but higher than in Garissa (~40-50%). At annual discharges between 3 and 4 km<sup>3</sup>, 60% of the TC in Garsen was accounted for by POC, while it was very variable at higher annual discharge. Both dissolved fractions showed the opposite pattern: the contribution of DOC in Garissa decreased from ca. 9% to 5%, while in Garsen it decreased initially between 8% and 4%, and became scattered at high annual water fluxes. The DIC was the most important C species at an annual discharge below 3 km<sup>3</sup>
- 25 yr<sup>-1</sup> at Garissa and below 2.75 km<sup>3</sup> yr<sup>-1</sup>at Garsen, but then decreased significantly. The scatter which was observed at high discharge in Garsen was related to the occurrence of flooded conditions, whereby the years with a high number of flooded days had a higher contribution of DIC and DOC.

#### **4** Discussion

## 4.1 Sampling frequency and timing

The added value of high frequency sampling in this river system will be demonstrated by analysis of the sediment sampling frequency. Two major effects can be derived from this analysis for a year without (2012) and with flooding (2013) (Figure

9). Firstly, the large range in annual fluxes at lower sampling frequencies indicates that when the sampling interval is above ca. 10 days, total annual fluxes cannot be reliably estimated. Secondly, calculated fluxes differ considerably if the different rating curves for flooded and non-flooded conditions were used (double curve) or not (single curve). In 2012, when no flooded conditions were observed, the annual flux was underestimated when all sampling points (including flooded points in

the dataset) were used for the construction of the rating curves. The opposite was true for the year 2013 when flooding occurred. This clearly shows that the identification of different regimes is important in order to correctly estimate annual fluxes.

When the sampling became less frequent, the number of samples taken under flooded conditions decreased rapidly and became very small when the sampling interval exceeded 7 days (Table 2). It may therefore be worthwhile to increase the

10 sampling frequency during the wet seasons, in order to be able to identify different regimes. Similar results were obtained in a small stream where additional measurements during high discharge events increased the accuracy of the DOC load significantly (Büttner and Tittel, 2013).

The results of the annual C flux calculations showed that the hydrological characteristics of the wet seasons in a specific year formed the dominant factor for the sediment and C flux estimates in that year. Years with a similar total annual discharge

- may still show important differences in sediment and C flux, as the discharge in one year can be well distributed over small peaks and not lead to significant flooding or a period of significant floods in combination with a prolonged period of low discharges may be experienced. This again illustrates that, for river systems with a high day-to-day variation in discharge, very high frequency measurements (e.g. daily) are a requirement for an accurate calculation of annual fluxes.
- Despite the uncertainty related to the inter-annual variability, our annual flux estimates showed clear tendencies as a function of annual discharge. As more data is collected, these trends can be further refined. In addition, the sampling effort can be made more efficient through the identification of the different hydrological regimes and the adaptation of the sampling frequency to the characteristics of the regime. Finally, a perfect rating curve is not possible because there will always be a considerable amount of scatter associated with the natural variability of the concentrations.

#### 4.2 Comparison with other measurements

- Previous annual flux estimates for the Tana River by Tamooh et al. (2014) were based on monthly samplings from 2009 to 2011 at Garissa (n=39) and TRPR (n=40), the latter located ca. 80 km upstream of Garsen (measured along the river). The annual fluxes were calculated using two software applications which use different calculation methods. GUMLEAF calculated discharge weighted average concentrations, while LOADEST fitted a regression line based on the (Adjusted) Maximum Likelihood Estimation (Tamooh et al., 2014).
- The annual fluxes of TSM over the period 2009-2011 at Garissa and TRPR were very comparable between the three methods (Table 3). This good fit can be explained by the absence of flooded days in this time period, whereas the flooded days in the extensive dataset are not included in the fit for the non-flooded regression line. Estimated annual fluxes of sediment and C in TRPR are much lower in comparison to the up- and downstream sites of Garissa and Garsen, except for

15

DIC. This is potentially due to the methodology of the discharge calculation at TRPR used by Tamooh et al. (2014), whereby the very high discharges were suppressed, leading to much lower discharge in TRPR compared to Garsen. However, even in our calculations, whereby the discharge in TRPR is calculated as a distance-weighted average of the discharge in Garissa and Garsen, the annual TSM flux in TRPR is less than the flux in Garissa and Garsen. This can be explained by significant

- 5 deposition of sediment between Garissa and TRPR and a subsequent mobilization of sediment between TRPR and Garsen. An alternative and more likely explanation is that the discharge at TRPR is underestimated by the distance-weighted average method, and as a result the TSM flux is also underestimated. Downstream of TRPR, the river shifts from a meandering river pattern to a more anastomosing pattern. The water loss is likely to be larger in the anastomosing part of the river and therefore the linear interpolation of discharge loss will lead to an underestimation of the discharges in TRPR. The
- 10 comparison of the average fluxes over the three year period with the average fluxes over the longer time frame indicates that 2009-2011 was a period with relatively low fluxes.

Comparisons between catchments are usually made based on the specific yields (SY), which is the flux normalized to the upstream catchment area (32 500 km<sup>2</sup>, 66 500 km<sup>2</sup> and 81 700 km<sup>2</sup> for the Tana River in Garissa, TRPR and Garsen respectively) (Table 2). The SY in Garissa is roughly three-fold the SY in Garsen. This can be explained by the very large catchment area between Garissa and Garsen which rarely delivers sediment to the river due to the semi-arid conditions.

- The specific sediment yield of African catchments ranges between 0.2 and 15 700 t km<sup>2</sup> yr<sup>-1</sup> with median and average values of 160 and 634 t km<sup>2</sup> yr<sup>-1</sup>, respectively (Vanmaercke et al., 2014). The values for the Tana River (203.3 and 46.0 t km<sup>2</sup> yr<sup>-1</sup> for Garissa and Garsen respectively) are well within this range, although on the lower side. Sediment yields decrease with increasing catchment area, and applying the regression equation between SY and catchment area developed by Vanmaercke
- et al. (2014) to our sites resulted in a predicted specific sediment yield of 79.2 and 68.3 t km<sup>2</sup> yr<sup>-1</sup> for Garissa and Garsen respectively. Considering the generalisations and assumptions, those predictions are fairly well in line with our measurments, with a gross underestimation for Garissa (203.3 t km<sup>2</sup> yr<sup>-1</sup>) and a small overestimation for Garsen (46.0 t km<sup>2</sup> yr<sup>-1</sup>) in respect to our calculations.

Comparing the C yields of the Tana River with other river systems indicated a relatively high POC yield in the Tana River

and in general a limited yield of dissolved C; A global dataset on SY for organic C indicates that the SY of POC ranged between 0.002 and 92.5 t km<sup>2</sup> yr<sup>-1</sup>, and between 0.001 and 56.9 t km<sup>2</sup> yr<sup>-1</sup> for DOC (Alvarez-Cobelas et al., 2012), which are very broad ranges which encapsulate our observations. The tropical rivers dataset compiled by Huang et al. (2012) calculated average yields for POC, DOC and DIC of 0.33, 1.00 and 0.63 t km<sup>2</sup> yr<sup>-1</sup>, respectively, for tropical Africa, and 2.05, 2.13 and 3.29 t km<sup>2</sup> yr<sup>-1</sup>, respectively, for the tropics in total.

## 30 4.3 Implications of environmental changes

The dependency of retention/mobilization on the total annual discharge implies that changes in the hydrological regime are expected to have a significant influence on both the sediment and C fluxes in the river, irrespective of whether those changes are due to climate change or human impact in the catchment. Projections for the precipitation in the area of the Tana River

show a tendency for increased precipitation (IPCC, 2013). This would result in higher annual discharge fluxes which in turn would lead to enhanced storage of sediment and retention of C between both sites (Figure 7).

The human impact due to the construction of the High Grand Falls dam is expected to act at two different levels: the sediment supply and the seasonal water distribution (Tana and Athi Rivers Development Authority, 2016). Earlier analyses

- 5 of the sediment fluxes at Garissa have shown that floodplain processes were able to buffer the sediment losses caused by the dams (Geeraert et al., 2015). If, after construction of the HGF dam, the sediment load of the river would not be at transport capacity anymore when it reaches Garissa, it can be expected that the degree of retention between Garissa and Garsen will decrease and the trendlines in Figure 7 will shift down as sediment is mobilized from the floodplain.
- The seasonal change in water discharge following the construction of the present cascade of dams can be summarized as a decrease of the discharge during the long wet season and an increase in discharge during the dry seasons (Maingi and Marsh, 2002). If this pattern would be enhanced after the construction of the HGF dam, then a larger fraction of the annual water flux would occur during the dry season. This is likely to result in a decrease of retention of sediment and C between both sites, as the dry-season flux of TSM and total C is larger in Garsen than in Garissa (Figure 5). The filling of the reservoir will also reduce the downstream water flux during the initial years as the volume of the reservoir (5.7 km<sup>3</sup>) is in the same order of
- magnitude as the measured total annual water flux in either 2012 or 2013 (Figure 5). Even after the filling of the dam, annual discharge is expected to decrease due to abstraction of water for irrigation purposes and also increased evaporation. In both situations, the management decisions with respect to the timing and magnitude of the water release will affect the downstream C flux.

#### **5** Conclusions

- The analysis of the water, sediment and C fluxes has revealed that the differences in annual fluxes are mainly determined by the characteristics of the wet season hydrograph, while the magnitude of the dry season fluxes is fairly constant. The sediment fluxes decreased strongly between Garissa and Garsen during the short wet and long wet seasons, while the annual TC flux showed only a slight decrease in the downstream direction. POC was the dominant C species over the whole year, but the DIC flux was slightly larger than the POC flux during some dry seasons. The DOC flux was always the smallest contributor to the TC flux and showed limited spatial and temporal variation.
  - A higher sampling frequency increased the accuracy of the flux estimates and led also to the identification of a different regime during flood periods, which are characterised by lower TSM and C concentrations. When flooded and non-flooded conditions are present in the sample database, the annual TSM flux of non-flooded years was underestimated when the differentiation between the regimes was not made, while the flux of flooded years was overestimated. Increasing the
- sampling frequency when shifts in the discharge regime are expected will therefore improve the annual flux estimates. Applying the current relationships between discharge and concentration revealed that at lower annual discharge, there was on average a downstream increase in sediment and C fluxes, while retention between both sites was expected at higher

annual discharge values. The integrated signal over the hydrological range was a net retention of TSM, POC, DOC as well as DIC. The relative fractions of the C species varied also as a function of the annual discharge. The DIC was the dominant C species at low annual discharge, while the POC dominated at high annual discharge. The relative importance of DOC was always below 10% and decreased as the annual discharge increased.

5 While we have to be careful with the extrapolation of C concentrations outside of the temporal range of observations, the trends above appear to be robust. Based on these trends, increased rainfall in the catchment due to climate change is expected to lead to increased retention of sediment and C between the Garissa and Garsen. The effect of the High Grand Falls dam will depend on management decisions concerning the regulation of the outlet discharge, but if discharges during the dry seasons are increased, this may lead to less retention of sediment and C in the floodplain.

## 10 Author contribution

S. Bouillon, N. Geeraert, T. Marwick, F.O. Omengo and F. Tamooh were involved in the data collection. N. Geeraert prepared the manuscript with contributions from all co-authors.

## **Competing interests**

The authors declare that they have no conflict of interest.

## 15 Acknowledgements

Funding was provided by the KU Leuven Special Research Fund, the Research Foundation Flanders (FWO–Vlaanderen, project G024012N), and an ERC Starting Grant (240002, AFRIVAL). We are grateful to the Kenya Wildlife Service (KWS) for assistance during field experiments and to the Water Resources Management Authority (WRMA) for assistance during the ADCP measurements and for sharing discharge data. AVB is a senior research associate at the FRS-FNRS.

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
