# Peer review of "Seasonal and inter-annual variations in carbon fluxes in a tropical river system (Tana River, Kenya)"

_Biogeosciences, 2017_

## Referee Comment (RC1) · C. Panagiotopoulos (Referee) · 15 Mar 2017

A. General comments This paper assesses the carbon (POC, DIC and DOC species) and sediment (TSM) fluxes of the Tana River (Kenya) from 2012 to 2014 in three distinct sides during wet and dry seasons. The authors provide results regarding the dynamics of the above species and their findings are extrapolated to the period 1942-2014. The Tana River has been well studied in previous investigations in terms of sediment mobilization (Geeraert et al., 2015), carbon dynamics (Geeraert et al., 2017), and organic carbon decomposition in relation with ïА̧d'13C isotopes (Geereart et al., 2016) from the same authors which makes this contribution important to the above findings. Nevertheless, I believe that an important part of the information included in this study is already

presented in 2017 (Biogeochemistry), notably all the part regarding carbon species flux, and the only new thing is the extrapolation of these data to the period 1942-2014. This in my opinion makes the scope of this manuscript too limited and as such the MS lacks of originality. A deeper scientific objective regarding the evolution of carbon fluxes in the future is lacking considering the anthropogenic pressures (as the authors state) in studied area. The present study provided indeed more accurate flux values but as far as I can see from Table 3, it is not clear if there is a significant difference for C-species measurements using the model or not (maybe additional tests are necessary or I missed something in the text?). Moreover, it is not clear if the constructions of dams (1960-1980s; according to Geeraert et al., 2015) are taken into account by the authors to validate their model when estimated past C-species flux. The present data are definitively publishable but not to a high ranked journal as Biogeosciences. I think continental shelf research would be a more appropriate journal for the manuscript. B. Specific comments Abstract: line 21. It is not clear from the text how variations in the discharge regime are related to climate changes, this is not explained further in the MS. Please elaborate. Study area: It is not clear if inundation events occur in the study area especially under high rainfall regimes (although I believe that this is improbable due to the canalization & dams) but the authors have to comment on this. This is important in order to understand if there are interactions between the river water and the surrounding floodplain when you calculate your fluxes. Long-term discharge dataset: It is not clear why the discharge break point was set at 500 m3 s-1?

---

## Referee Comment (RC2) · Anonymous Referee #2 · 2 May 2017

REVIEW Seasonal and inter-annual variations in carbon fluxes in a tropical river system (Tana River, Kenya)

SUMMARY This paper studies the dynamics of water, carbon and sediment transport within the Tana River using data at three sites from 2012 to 2014. The major findings from the study are that the characteristics of the wet season hydrograph dominate the quantification of these fluxes while the dry season discharge is fairly consistent. Another conclusion from the study is that higher sampling frequency is needed to accurately estimate the carbon and sediment fluxes.

GENERAL COMMENTS I have several major concerns with this study. First, it seems that a lot of this data has already been published elsewhere. The authors fail to describe how the current manuscript is different from previous studies done on the Tana River by the same research group or a subset of this group.

Secondly, a majority of the conclusions were made using 2 years of high temporal resolution data that had different hydrological regimes. So, is it not surprising that the results indicate that the majority of the difference is associated with hydrological regimes? In my opinion, a lot more can be done with the data that is available. Why not look at concentration temperature relationships? It is not clear how large or small temperature fluctuations at the site were in terms of both seasonal and annual trends.

The authors also simplify all the assumptions regarding retention or mobilization of carbon. There could be a strong impact of microbial reactions on POC, DOC and DIC fluxes. Note that microbially mediated breakdown of DOC can result in a pH decrease accompanied by an increase in bicarbonate alkalinity. Thus, DIC and DOC fluxes can be interlinked. How do the authors address this linkage in understanding patterns of DIC and DOC fluxes along the Tana river?

The publication is also missing recent references that are very much relevant to the current study. For example, Arora et al. (2016, Biogeochemistry); Raymond et al. (2013, Nature); and Van Cappellen and Maavara (2016, Ecohydrology & Hydrobiology).

SPECIFIC COMMENTS

Page 5 Para 20 Slightly more detail can be added to the sentence stating the differences in sampling protocols, especially differences in DIC collection methods.

Page 5 Para 25 A reference should be provided for the maximum POC concentration chosen for this study.

Page 6 Para 25 "dry seasons still had a fair share" of what?

---

## Author Comment (AC1) · 2 May 2017

We thank C. Panagiotopoulos for the review of the paper that help us clarify the objectives of the paper and some points of the discussion.

General comments

RC: This paper assesses the carbon (POC, DIC and DOC species) and sediment (TSM) fluxes of the Tana River (Kenya) from 2012 to 2014 in three distinct sides during wet and dry seasons. The authors provide results regarding the dynamics of the above species and their findings are extrapolated to the period 1942-2014. The Tana River has been well studied in previous investigations in terms of sediment mobilization

(Geeraert et al., 2015), carbon dynamics (Geeraert et al., 2017), and organic carbon decomposition in relation with d13C isotopes (Geereart et al., 2016) from the same authors which makes this contribution important to the above findings. Nevertheless, I believe that an important part of the information included in this study is already presented in 2017 (Biogeochemistry), notably all the part regarding carbon species flux, and the only new thing is the extrapolation of these data to the period 1942-2014. This in my opinion makes the scope of this manuscript too limited and as such the MS lacks of originality.

REPLY: This dataset used in this manuscript is indeed largely the same as our Biogeochemistry (2017) paper, but includes some additional data from previous research within our research group. The scope of the manuscript is, however, entirely different and what we present here could not be included in the 2017 Biogeochemistry paper as this would have made the discussion too complex and because the scope of the new paper is very different. While our previous work discussed in detail the C dynamics during three specific campaigns, here we explore the consequences of our observations (different dynamics during flooded and non-flooded seasons) on the riverine carbon fluxes over a longer time scales. There is, in our opinion, no overlap in the conclusions or take-home message of both papers.

RC: A deeper scientific objective regarding the evolution of carbon fluxes in the future is lacking considering the anthropogenic pressures (as the authors state) in studied area.

REPLY: The different parts of this manuscript are building blocks to better assess how C fluxes may change due to anthropogenic pressure. First, we want to point out that no clear contrasts between pre-dam and post-dam sediment concentrations were found, but that major differences are observed depending on whether high discharges occurred with flooding or without flooding (Geeraert et al. 2015). We referred to this as the different hydrological regimes. So for the analysis of the C dynamics, we continued focussing on the different regimes.
Therefore, in section 3.1 and 3.2, we compare the year 2012 and 2013, because the former is an example of a year without flooding, while 2013 experienced severe flooding. Analysing the seasonal contrasts between those two years provides insights which are useful to predict the effects of human induced changes that would alter the seasonal variation of the discharge (e.g. through the implementation of a different scheme for dam releases). We use the longitudinal variation to assess how retention or release of sediment and C will change when the frequency of flooded years would change. Evidently, such predictions depend on the accuracy of the data available. To assess the potential impact of errors we would make when not including the differentiation between those two hydrological conditions, we investigated the impact of sampling frequency and timing on data quality and predictions (section 4.1).

In section 4.3, we bring all this information together with the expected hydrological changes due to dam construction and climate change so that potential future changes can be assessed.

RC: The present study provided indeed more accurate flux values but as far as I can see from Table 3, it is not clear if there is a significant difference for C-species measurements using the model or not (maybe additional tests are necessary or I missed something in the text?).

REPLY: The good fit of the data for the period 2009-2011 is due to the fact that there was only one hydrological regime (non-flooded) during that period. If the same method would have been used for the period 2009-2013 (with the measurements from a flooded season included), we would have ended up with an underestimation of the annual fluxes in the non-flooded years (2009-2012) and an overestimation in 2013. This is represented in Figure 9 as the "Single curve" compared to the "Double curve". This is explained on page 9, l.30-31, but maybe too briefly to be fully understandable. We will therefore adjust our explanations to make this more understandable.

RC: Moreover, it is not clear if the constructions of dams (1960-1980s; according to

Geeraert et al., 2015) are taken into account by the authors to validate their model when estimated past C-species flux.

REPLY: First, we did not intend to reconstruct the past fluxes, we simply used the historical hydrological record to represent annual patterns in discharge as well as variations in the frequency of high discharge events and used this for our simulations.

There are no empirical data available that would allow to directly estimate C fluxes before the construction of the dams. However, we found no clear differences in the sediment fluxes before and after dam construction (fig. 4, Geeraert et al., 2015), while the pre-dam period also experienced contrasts between flooded and non-flooded conditions. Particulate OC is a relatively constant fraction of the sediment flux and is the major OC fraction transported by the river: this suggests that the conclusion with respect to the impact of different flooding regimes on C and sediment transfer will hold for pre-and post-dam OC fluxes.

RC: The present data are definitively publishable but not to a high ranked journal as Biogeosciences. I think continental shelf research would be a more appropriate journal for the manuscript.

Specific comments

RC: Abstract: line 21. It is not clear from the text how variations in the discharge regime are related to climate changes, this is not explained further in the MS. Please elaborate.

REPLY: In section 4.3 (p10 L.33 –p11 L. 1), we refer to the IPCC projections for the area which show a tendency for increased precipitation, which would subsequently lead to higher annual discharge fluxes.

RC: Study area: It is not clear if inundation events occur in the study area especially under high rainfall regimes (although I believe that this is improbable due to the canalization & dams) but the authors have to comment on this. This is important in order

to understand if there are interactions between the river water and the surrounding floodplain when you calculate your fluxes.

REPLY: There is indeed inundation/flooding taking place, as is explained in section 2.2 when the river experiences considerable flooding in the floodplain. High rainfall is extremely rare in the lower Tana valley (referred to in the manuscript as semi-arid environment without tributaries), while the dams are located hundreds of kilometres upstream of our research area. When looking at the fluxes in the river itself, the interaction with the floodplain is present, but it doesn't affect the interpretation of the annual fluxes. The interaction with the floodplain has been examined in more detail in Geeraert et al. (2017, Biogeochemistry).

RC: Long-term discharge dataset: It is not clear why the discharge break point was set at 500 m3 s-1?

REPLY: The break point was set at 550 m3s-1, because that worked best during our initial calculations of sediment fluxes in Geeraert et al. (2015). We decided to use the same value for consistency.

---

## Author Comment (AC2) · 11 May 2017

We thank Anonymous Referee #2 for her/his comments on the paper that help us clarify the objectives of the paper and some points of the discussion.

GENERAL COMMENTS RC: I have several major concerns with this study. First, it seems that a lot of this data has already been published elsewhere. The authors fail to describe how the current manuscript is different from previous studies done on the Tana River by the same research group or a subset of this group.

REPLY: The dataset which is used for this analysis is indeed previously published (primarily in Geeraert et al. 2017, Biogeochemistry). In that article, we focussed on the details of the C dynamics during three wet season campaigns. In the current manuscript, we want to use the detailed dataset to gain more insight in the long-term C fluxes in the river system and the potential effects of hydrological changes due to climate change or dam management. We acknowledge that this objective was not clearly explained in the manuscript and will rewrite it for clarification.

RC: Secondly, a majority of the conclusions were made using 2 years of high temporal resolution data that had different hydrological regimes. So, is it not surprising that the results indicate that the majority of the difference is associated with hydrological regimes?

REPLY: The identification of the different hydrological regimes was already presented in previous studies (Geeraert et al. 2015, Geeraert et al. 2017) and we used this knowledge here to further examine how that could affect the analysis and interpretation of annual C fluxes. The differences in seasonal variations in the two regimes were analysed in section 3.1 and 3.2, while in section 4.1, we presented what the error on the annual flux would be if we would fail to recognise the different hydrological regimes. Those insights are needed to consider future changes in fluxes due to changes in hydrology. It important to consider that the effects of different hydrological regimes can be mechanistically explained, i.e. we explain the observed differences between flooded and non-flooded high flows and the dry season by looking at the processes controlling carbon and sediment dynamics. Thus, we are not just using the difference between the various regimes in a statistical sense. Evidently, the quantification of the different effects is characterised by a large uncertainty: but this uncertainty is explicitly accounted for in our analysis.

RC: In my opinion, a lot more can be done with the data that is available. Why not look at concentration temperature relationships? It is not clear how large or small temperature fluctuations at the site were in terms of both seasonal and annual trends.

REPLY: The spatial and temporal variation of the physico-chemical parameters was

examined in our Biogeochemistry paper (Geeraert et al. 2017) and no significant relationships with temperature were observed which could impact the C fluxes. Therefore, they were not further discussed in this manuscript. Temperature variations were very limited, 27.0+-1.5 °C and 28.5 +/- 1.7 °C in Garissa and Garsen respectively over all of our campaigns.

RC: The authors also simplify all the assumptions regarding retention or mobilization of carbon. There could be a strong impact of microbial reactions on POC, DOC and DIC fluxes. Note that microbially mediated breakdown of DOC can result in a pH decrease accompanied by an increase in bicarbonate alkalinity. Thus, DIC and DOC fluxes can be interlinked. How do the authors address this linkage in understanding patterns of DIC and DOC fluxes along the Tana river?

REPLY: The linkages and interactions between the different C species are discussed in depth in our Biogeochemistry (2017) paper. The fluxes from one C pool in the river to another were based on measurements of respiration rates and pCO2 and by closing the C budget of the river system. There are much more assumptions involved in these calculations and therefore, we didn't want to extrapolate the calculation of those fluxes outside of the observation time frame. The measurement of the concentrations are more robust and are suitable to expand in time.

RC: The publication is also missing recent references that are very much relevant to the current study. For example, Arora et al. (2016, Biogeochemistry); Raymond et al. (2013, Nature); and Van Cappellen and Maavara (2016, Ecohydrology & Hydrobiology).

REPLY: We included the references of Raymond et al. (2013) and Van Cappellen and Maavara (2016), but didn't include the one of Arora et al. (2016) because it was focussing on the C processes in soils, while our focus is on the magnitude of the different C fluxes, and how they would change when a change in discharge occurs in the river.

SPECIFIC COMMENTS RC: Page 5 Para 20 Slightly more detail can be added to the

sentence stating the differences in sampling protocols, especially differences in DIC collection methods.

REPLY (option 1): We reformulated this sentence stating that an empirical relationship with TA was used for the calculation of the DIC.

RC: Page 5 Para 25 A reference should be provided for the maximum POC concentration chosen for this study.

REPLY: This value was chosen as an informed estimate (a reasonable value as the average of the three highest observations) to correct for values which were unrealistic as a result of the exponential shape of the regression equation. In addition, the number of days for which that correction was needed, was very limited; the correction was, depending on the regression line (figure 4), needed between 93 and 1128 times over the nearly 27 000 days in Garissa and between 135 and 1409 times in Garsen.

RC: Page 6 Para 25 "dry seasons still had a fair share" of what?

REPLY: They had a fair share in the total annual discharge (34-44%). For clarity, we rephrased the sentence to provide a better contrast with their proportion in time: The dry seasons still accounted for at least on third of the total annual discharge (34-44%), which is considerably smaller than their proportion in time ($\sim$58% of the year).